# MagNet: Multi-Level Attention Graph Network for Predicting High-Resolution Spatial Transcriptomics

**Junchao Zhu**[1]                          JUNCHAO.ZHU@VANDERBILT.EDU
**Ruining Deng**[2]                          RUD4004@MED.CORNELL.EDU
**Tianyuan Yao**[1]                          TIANYUAN.YAO@VANDERBILT.EDU
**Juming Xiong**[1]                          JUMING.XIONG@VANDERBILT.EDU
**Chongyu Qu**[1]                          CHONGYU.QU@VANDERBILT.EDU
**Junlin Guo**[1]                          JUNLIN.GUO@VANDERBILT.EDU
**Siqi Lu**[1]                          SIQI.LU@VANDERBILT.EDU
**Yucheng Tang**[4]                          YUCHENGT@NVIDIA.COM
**Daguang Xu**[4]                          DAGUANGX@NVIDIA.COM
**Mengmeng Yin**[3]                          MENGMENG.YIN.1@VUMC.ORG
**Yu Wang**[3]                          YU.WANG.2@VUMC.ORG
**Shilin Zhao**[3]                          SHILIN.ZHAO.1@VUMC.ORG
**Yaohong Wang**[5]                          YAOHONGWANG@MDANDERSON.ORG
**Haichun Yang**[3]                          HAICHUN.YANG@VUMC.ORG
**Yuankai Huo**[1]                          YUANKAI.HUO@VANDERBILT.EDU

[1] *Vanderbilt University, TN, USA*

[2] *Weill Cornell Medicine, NY, USA*

[3] *Vanderbilt University Medical Center, TN, USA*

[4] *NVIDIA, CA, USA*

[5] *UT MD Anderson Cancer Center, TX, USA*

**Editors:** Accepted for publication at MIDL 2025

## Abstract

The rapid development of spatial transcriptomics (ST) offers new opportunities to explore the gene expression patterns within the spatial microenvironment. Current research integrates pathological images to infer gene expression, addressing the high costs and time-consuming processes to generate spatial transcriptomics data. However, as spatial transcriptomics resolution continues to improve, existing methods remain primarily focused on gene expression prediction at low-resolution (55 $\mu$m) spot levels. These methods face significant challenges, especially the information bottleneck, when they are applied to high-resolution (8 $\mu$m) Visium HD data. To bridge this gap, this paper introduces MagNet, a multi-level attention graph network designed for the accurate prediction of high-resolution HD data. MagNet employs cross-attention layers to integrate features from multi-resolution image patches hierarchically and utilizes a GAT-Transformer module to aggregate neighborhood information. By integrating multilevel features, MagNet overcomes the limitations posed by low-resolution inputs in predicting high-resolution gene expression. We systematically evaluated MagNet and existing ST prediction models on both a private spatial transcriptomics dataset and a public dataset at three different resolution levels. The results demonstrate that MagNet achieves state-of-the-art performance at both spot level and high-resolution bin levels, providing a novel methodology and benchmark for future research

and applications in high-resolution HD-level spatial transcriptomics. Code is available at https://github.com/Junchao-Zhu/MagNet.

**Keywords:** Spatial Transcriptomics, Computational Pathology, Medical Image Analysis

## 1. Introduction

Spatial transcriptomics (ST) provides a novel view for correlating pathological tissue structures with their spatial gene expression patterns (Burgess, 2019; Asp et al., 2019; He et al., 2020; Zhu et al., 2024). This approach advances the development of effective treatment strategies (Asp et al., 2020). Studies have demonstrated a strong correlation between features of pathological images and their gene expression patterns (Badea and Stănescu, 2020). Such findings have motivated the development of image-based methods for predicting gene expression, offering a non-destructive and cost-effective alternative to traditional sequencing techniques.

In recent years, the widespread application of deep learning methods in medical image analysis (Ke et al., 2023; Zhu et al., 2023; Qu et al., 2025) has provided multiple useful tools. These methods have facilitated the integration of pathology images with other data modalities by automating image interpretation processes (Deng et al., 2025; Zhu et al., 2025). Currently, several studies have employed methods such as convolutional neural networks (CNNs) (He et al., 2020; Yang et al., 2023) and graph neural networks (GNNs) (Pang et al., 2021; Zeng et al., 2022; Jia et al., 2024) to predict spatial transcriptomic expression at the spot level with low resolution. These approaches exploit spatial dependencies (Zeng et al., 2022; Pang et al., 2021) and image similarities (Xie et al., 2024; Yang et al., 2023) inherent in pathological images, thus integrating information to optimize the fusion of image features. Such advances address the challenges of scarce high-quality spatial transcriptomic data and the high cost of acquisition.

Continuous advancements in ST sequencing technology (Ståhl et al., 2016; Wang et al., 2018; Eng et al., 2019) have significantly improved the resolution of existing ST data, as is shown in Figure 1, which has progressed from the initial 55 $\mu$m spots to higher resolutions, such as Visium HD data with bin diameters of 8 $\mu$m or even 2 $\mu$m. Such advancement enables a more comprehensive analysis of the relationship between pathological tissues and gene expression at the single-cell level (Benjamin et al., 2024; Oliveira et al., 2024; Janesick et al., 2023). However, current deep-learning methods face an information bottleneck when dealing with high-resolution HD data (Tishby and Zaslavsky, 2015). Specifically, the limited information from low-resolution input images is insufficient to effectively support the prediction of high-dimensional gene expression. The features extracted by these models may lack the complexity required to represent the intricate details of high-resolution, high-dimensional gene expression data.

To address this issue, this paper proposes MagNet, a Multi-Level Attention Graph Network designed for accurate prediction of high-resolution HD data. MagNet integrates information across multiple resolutions, including the bin, spot, and region levels, through cross-attention layers. MagNet also extracts and combines features from neighboring regions with Graph Attention Network (GAT) and Transformer layers. Thus, our proposed framework overcomes the information bottleneck posed by low-resolution inputs when predicting high-resolution, high-dimensional gene expression by efficient extraction and integration of

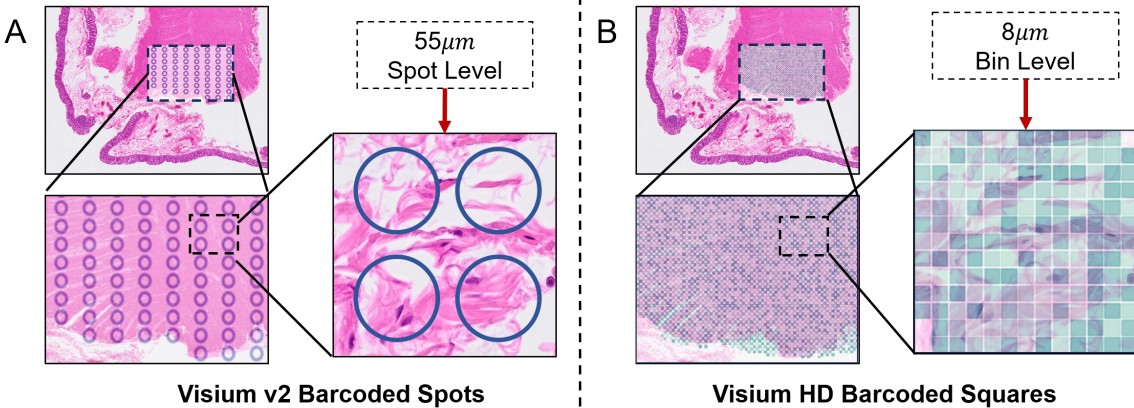

Figure 1: **Spatial transcriptomics data at different resolutions.** (A) Traditional low-resolution 10X Visium v2 barcoded spots, where spots are discretely distributed with a diameter of 55 $\mu$m. (B) Current high-resolution 10X Visium HD barcoded squares, where bins are densely distributed with a diameter of 8 $\mu$m.

multisource and multilevel features. Furthermore, the model incorporates cross-resolution constraints on gene expression within the same region, further enhancing its performance in HD gene expression prediction. Our contributions can be summarized in three aspects:

- We present MagNet, a Multi-Level Attention Graph Network designed for accurate prediction of high-resolution HD data. To our knowledge, it is the first model dedicated to HD-level gene expression prediction.
- Our proposed framework leverages cross-attention layers and GAT-Transformer blocks to effectively extract and integrate multi-source and multi-level features, tackling the information bottleneck of low-resolution inputs in predicting high-resolution ST expression.
- We provide our model as an open-source tool, benchmarking and providing a systematic evaluation on a privately-collected kidney HD ST dataset and a public colorectal cancer HD ST dataset.

## 2. Method

### 2.1. Unified Cross-Resolution Feature Aggregation

We cropped patches at the bin, spot, and region levels for each bin $i$, denoted as $i_b$, $i_s$ and $i_r$. Features of these patches, represented as $f_b$, $f_s$ and $f_r$, are extracted by a pre-trained ResNet50 (He et al., 2016). We adopt the strategy proposed by TRIPLEX (Chung et al., 2024) that freezes the encoder parameters for the spot and region levels while updating only the bin-level encoder to minimize computational overhead.

To refine the representation of $f_b$, the features of other resolutions are treated as the key matrix (K) and the value matrix (V), with $f_b$ acting as the query matrix (Q). A cross-attention layer is used to effectively merge the features of $f_s$ and $f_r$ into $f_b$. Thus, the fused

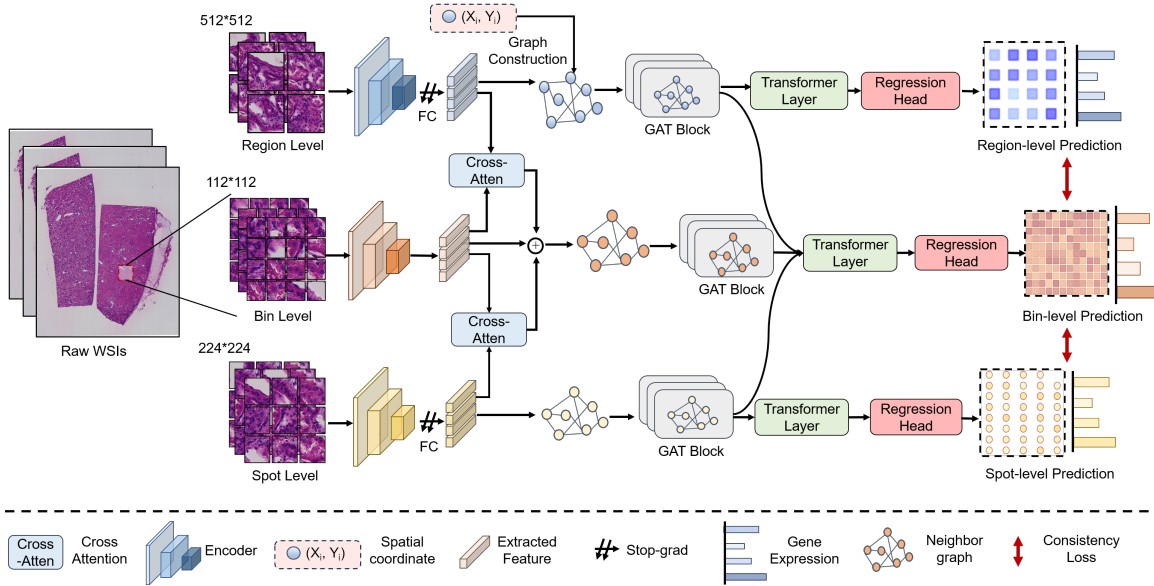

Figure 2: **The network structure of the proposed MagNet.** MagNet utilizes cross-attention layers to integrate features extracted from multi-resolution patches. Additionally, it incorporates a GAT-Transformer block to aggregate neighborhood information while leveraging spatial relationships. The predictions for each resolution level are then independently generated by a regression head.

feature $f'_b$ is formulated as:

$$f'_b = \text{softmax}\left(\frac{f_b f_i^T}{\sqrt{d}}\right) f_i, \quad i = s, r \tag{1}$$

where $\sqrt{d}$ is a scaling factor. Finally, by concatenating the features from all three levels, the fused multi-level feature $F$ is obtained for use in subsequent processes.

## 2.2. Spatial-Guided Graph Integration Block

To exploit the spatial relationship of pathological images, we propose a spatially-guided graph integration block that integrates GAT and transformer layers. The connections between bins are first established by calculating the weight $e_{ij}$ between any two nodes $i$ and $j$ using the Euclidean distance. The top-$k$ lowest $e_{ij}$ values are selected to establish connections within the whole-slide image. The constructed graph is then fed into the spatial-guided graph integration block for further processing.

Subsequently, after rounds of graph attention convolution, the processed feature $F_m^i$ for each $i_b$, $i_s$ and $i_r$ is formulated as follows:

$$\mathbf{F}_m^i = \Big\|_{k=1}^{K} \sigma\left(\sum_{j\in\mathcal{N}(i)} \alpha_{ij}^k \mathbf{W}^k \mathbf{f}_m^j\right), \{m|b,s,r\} \tag{2}$$

where $\mathcal{N}(i)$ denotes the set of adjacent nodes, $\parallel$ represents concatenation operation, $\sigma$ is the activation function, $\alpha_{ij}^k$ is the weight of the $k$-th attention head, and $\mathbf{W}^k$ is a linear transformation matrix determined by the connections between nodes.

A Transformer layer is used for adaptive aggregation of neighborhood information from each round, thus enhancing the representation of features. Finally, the regression head generates gene expression predictions for each level separately, denoted as $p_b$, $p_s$, and $p_r$.

## 2.3. Loss Function

To exploit the mutual consistency among multilevel information, we designed a hybrid loss function comprising prediction loss $L_p$ and consistency loss $L_c$ to optimize the model learning process. The prediction loss primarily focuses on minimizing the discrepancies between the model's predictions and the ground truth at each resolution level. For the prediction task at bin level, we employ Mean Squared Error (MSE) and Pearson Correlation Coefficient loss (PCC) to evaluate the model's performance. To avoid introducing additional noise, only PCC loss is utilized to assess the model's performance at the spot and region levels. Hence, the prediction loss is formulated as:

$$L_p = MSE(p_b, y_b) + \sum_{i=b,s,r} \lambda_i \cdot PCC(p_i, y_i) \tag{3}$$

Here, $b$, $s$, and $r$ represent the bin, spot, and region levels, respectively. $p_i$ and $y_i$ denote the prediction of the model and its corresponding ground truth, while $\lambda_i$ is a hyperparameter used to balance the PCC loss at different resolution levels.

Since patches at different resolutions within the same region exhibit similar trends in gene expression, we employ PCC loss to constrain the differences between bin-level predictions and those at other levels. The consistency loss $L_c$ is defined as:

$$L_c = \lambda_1 \cdot PCC(p_b, p_s) + \lambda_2 \cdot PCC(p_b, p_r) \tag{4}$$

Thus, the overall loss of the model $L$ is defined as:

$$L = \gamma_1 \cdot L_p + \gamma_2 \cdot L_c \tag{5}$$

Here, $\gamma_1$ and $\gamma_2$ are hyperparameters used to balance the two types of losses, and they are set to 1 and 0.25 in the subsequent experiments.

## 3. Data and Experiment

**Dataset.** We benchmarked our MagNet and other baseline models on a privately collected kidney pathology dataset (VUMC) and a publicly available colorectal cancer (CRC) dataset (Oliveira et al., 2024). We conducted four-fold cross-validation at the WSI level. Our in-house dataset contains 12 HD ST samples with three resolutions: 2 $\mu$m, 8 $\mu$m, and 16 $\mu$m, where 1px in the WSI corresponds to 0.25 $\mu$m of real tissue. The CRC dataset consists of four samples with a single-layer section, including two CRC tissues and two adjacent normal tissues. The process has been approved by Institutional Review Board (IRB).

Table 1: **Quantitative comparisons across different datasets.** The best performance is highlighted in **bold**, where we can observe that `MagNet` outperforms the state-of-the-art in multiple resolutions.

| Resolution | Model | VUMC (in-house dataset) | | | CRC (Oliveira et al., 2024) | | |
|---|---|---|---|---|---|---|---|
| | | MSE | MAE | PCC | MSE | MAE | PCC |
| 8um/112px | ST-Net | 0.193±0.004 | 0.388±0.009 | 0.226±0.040 | 0.292±0.076 | 0.402±0.084 | 0.527±0.155 |
| | EGN | 0.048±0.011 | 0.134±0.020 | 0.157±0.024 | 0.409±0.164 | 0.508±0.139 | 0.511±0.152 |
| | HisToGene | 0.105±0.007 | 0.241±0.006 | 0.109±0.018 | 0.311±0.088 | 0.419±0.075 | 0.451±0.128 |
| | BLEEP | 0.063±0.006 | 0.163±0.009 | 0.199±0.052 | 0.348±0.041 | 0.440±0.0361 | 0.475±0.1379 |
| | His2ST | 0.140±0.019 | 0.358±0.026 | 0.175±0.033 | 0.287±0.113 | 0.4041±0.109 | 0.537±0.165 |
| | TRIPLEX | 0.151±0.152 | 0.286±0.180 | 0.107±0.059 | 0.291±0.110 | 0.397±0.069 | 0.498±0.167 |
| | MagNet (Ours) | **0.048±0.008** | **0.109±0.008** | **0.278±0.042** | **0.271±0.054** | **0.375±0.053** | **0.541±0.167** |
| 16um/112px | ST-Net | 0.288±0.007 | 0.420±0.027 | 0.364±0.0539 | 0.661±0.239 | 0.632±0.146 | 0.560±0.151 |
| | EGN | 0.149±0.037 | 0.302±0.06 | 0.308±0.037 | 0.740±0.0241 | 0.677±0.013 | 0.552±0.014 |
| | HisToGene | 0.204±0.045 | 0.380±0.052 | 0.243±0.035 | 0.660±0.176 | 0.6368±0.099 | 0.522±0.136 |
| | BLEEP | 0.174±0.029 | 0.290±0.031 | 0.317±0.058 | 0.673±0.161 | 0.625±0.088 | 0.504±0.123 |
| | His2ST | 0.224±0.044 | 0.427±0.049 | 0.330±0.046 | 0.610±0.168 | 0.611±0.103 | 0.562±0.152 |
| | TRIPLEX | 0.211±0.079 | 0.331±0.089 | 0.310±0.079 | 0.632±0.123 | 0.618±0.080 | 0.412±0.134 |
| | MagNet (Ours) | **0.127±0.024** | **0.228±0.034** | **0.378±0.057** | **0.564±0.184** | **0.581±0.114** | **0.574±0.154** |
| 55um/224px | ST-Net | 0.442±0.036 | 0.549±0.019 | 0.609±0.059 | 0.767±0.203 | 0.652±0.086 | 0.649±0.080 |
| | EGN | 0.355±0.030 | 0.471±0.010 | 0.601±0.0561 | 0.778±0.229 | 0.651±0.105 | 0.674±0.071 |
| | HisToGene | 0.403±0.028 | 0.517±0.017 | 0.596±0.058 | 0.702±0.173 | 0.622±0.074 | 0.663±0.067 |
| | BLEEP | 0.339±0.026 | 0.467±0.017 | 0.576±0.049 | 0.717±0.112 | 0.623±0.044 | 0.667±0.043 |
| | His2ST | 0.327±0.021 | 0.459±0.013 | 0.601±0.058 | 0.813±0.199 | 0.673±0.089 | 0.673±0.065 |
| | TRIPLEX | 0.442±0.200 | 0.525±0.119 | 0.579±0.075 | 0.828±0.148 | 0.688±0.048 | **0.677±0.059** |
| | MagNet (Ours) | **0.324±0.044** | **0.458±0.030** | **0.611±0.082** | **0.688±0.149** | **0.612±0.069** | 0.670±0.059 |

**Data Preprocessing.** 6,000 bins were randomly selected for each WSI, and 112×112 pixel patches centered at 8 $\mu$m and 16 $\mu$m bins were cropped. At the spot and region levels, patches with diameters of 224 and 512 pixels were extracted across the WSI, with their gene expressions aggregated from bin-level data. 2,500 spot-level patches per WSI were selected for training and testing. Patch pairing across levels was based on the distance between the coordinates in different resolutions. We follow the method proposed in ST-Net (He et al., 2020) and select the top 250 genes with the highest average expression levels of more than 20,000 original genes for prediction. Gene expression values were normalized using the approach introduced in TRIPLEX (Chung et al., 2024), which involves proportional normalization followed by a log transformation.

**Compared Methods and Evaluation Metrics.** MagNet was benchmarked against current ST counterparts, including multi-resolution-based network (Chung et al., 2024), spatial-aware methods HisToGene (Pang et al., 2021) and His2ST (Zeng et al., 2022), similarity-based strategy BLEEP (Xie et al., 2024), and EGN (Yang et al., 2023), and the classic approach ST-Net (He et al., 2020). We used the officially released code published along with the papers for all of the methods. The Pearson correlation coefficient (PCC), mean squared error (MSE), and mean absolute error (MAE) are used to evaluate the performance of the models comprehensively.

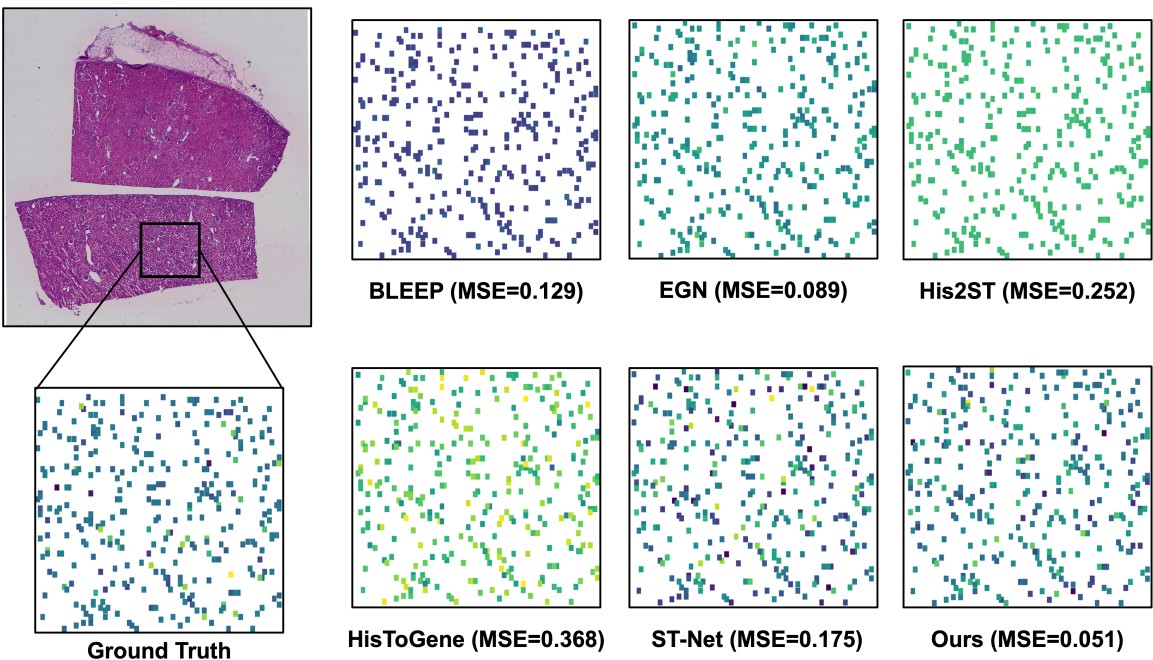

Figure 3: **Qualitative comparison for pivotal SGPP1 gene expression prediction.** SGPP1 expression prediction distribution of randomly selected $16\mu m$ bins within a region in WSI.

**Experiment Setting and Implementation.** Experiments were conducted on NVIDIA RTX A6000 GPU cards. The SGD optimizer was utilized, with momentum set to 0.9 and a weight decay of $10^{-4}$. An initial learning rate of $10^{-4}$ was applied, which followed a cosine decay schedule, decreasing it progressively to 1% of its initial value during training. All models are trained to converge. We employed a batch size of 256 for training and fine-tuned the hyperparameters $\lambda_1$, $\lambda_2$, $\lambda_b$, $\lambda_s$, and $\lambda_r$ in our hybrid loss function to values of 0.1, 0.1, 0.8, 0.25, and 0.25, respectively. For graph construction, the top-$k$ value was fixed at 8. We select 8 $\mu m$ and 16 $\mu m$ bins as the target HD resolution to predict, due to the extremely low gene expression amount in 2 $\mu m$ bins. During spot-level experiments, we freeze the encoder parameters of the bin and region levels and update the spot level instead.

## 4. Results

### 4.1. Cross-Validation Evaluation

We conducted four-fold cross-validation on the WSI level to validate and benchmark Mag-Net and SOTAs on the two HD datasets. Table 1 summarizes quantitative comparisons of various baselines across different datasets and resolutions. Our proposed MagNet consistently outperforms existing methods in almost all metrics, with its superiority particularly evident at HD high-resolution levels. Taking the 8 $\mu m$ prediction task in our VUMC dataset as an example, MagNet achieved MSE, MAE, and PCC values of 0.048±0.008, 0.109±0.008,

and 0.278±0.042, respectively, significantly surpassing the results of other methods, such as BLEEP, which reported values of 0.063±0.006, 0.163±0.009, and 0.199±0.052.

These findings demonstrate the capability of MagNet to effectively address the information bottleneck inherent in high-resolution gene prediction tasks. By efficiently integrating and leveraging multi-source and multi-level information, MagNet overcomes the performance limitations caused by constrained data and substantially enhances prediction accuracy for high-resolution HD data. Furthermore, the relatively low standard deviation observed among all metrics during cross-validation highlights the method's robustness and stability, underscoring its reliability for practical clinical applications.

Table 2: **Ablation study for functional blocks in MagNet.** The benefits from each designed block are orthonormal, while MagNet achieves optimal results when integrating all modules.

| Functional Blocks | VUMC (in-house dataset) /16$\mu$m | | | CRC (Oliveira et al., 2024)/16 $\mu$m | | |
|---|---|---|---|---|---|---|
| | MSE | MAE | PCC | MSE | MAE | PCC |
| w.o. GAT & Multi-resolution | 0.148±0.042 | 0.281±0.069 | 0.299±0.028 | 0.799±0.259 | 0.709±0.146 | 0.548±0.146 |
| w.o. GAT block | 0.135±0.030 | 0.266±0.048 | 0.306±0.043 | 0.632±0.170 | 0.624±0.096 | 0.550±0.147 |
| w.o. Multi-resolution | 0.133±0.030 | 0.260±0.051 | 0.323±0.044 | 0.634±0.175 | 0.628±0.111 | 0.563±0.152 |
| w.o. Consistency Loss | 0.130±0.023 | 0.235±0.040 | 0.369±0.054 | 0.624±0.187 | 0.619±0.117 | 0.559±0.146 |
| w. All blocks | **0.127±0.024** | **0.228±0.034** | **0.378±0.057** | **0.564±0.184** | **0.581±0.114** | **0.574±0.154** |

## 4.2. Pivotal Gene Expression Prediction

We evaluated the clinical applicability of various baselines by analyzing the predictive performance of key biomarker SGPP1 and tubule-related gene DPEP1 in our kidney dataset at 16$\mu$m level. SGPP1 and DPEP1 with their associated pathways play a critical role in kidney health and disease, with direct implications for conditions such as acute kidney injury and fibrotic kidney diseases (Drexler et al., 2021; Keller et al., 2024; Lovric et al., 2017).

Figure 3 illustrates the predictive performance of different models for the SGPP1 gene. Compared with other baseline models, our proposed MagNet achieved the best MSE of 0.051. Additionally, we analyzed DPEP1 and SGPP1 predictions on WSIs from two samples in our VUMC dataset. Results show that MagNet achieved MSEs of 0.0544 / 0.0493 for SGPP1 / DPEP1 at the WSI level, significantly outperforming other methods like EGN (0.1605 / 0.1855) and BLEEP (0.1530 / 0.1126), further validating its superiority in HD-level gene expression prediction. By deeply integrating and leveraging multi-level information, MagNet captures the spatial distribution of key gene expressions in pathological tissues with higher resolution.

Table 3: **Ablation study on high-resolution-level-only baseline.**

| Functional Blocks | VUMC (in-house dataset)/8$\mu$m | | | CRC (Oliveira et al., 2024)/8$\mu$m | | |
|---|---|---|---|---|---|---|
| | MSE | MAE | PCC | MSE | MAE | PCC |
| w.o. GAT blocks Multi-resolution | 0.052±0.023 | 0.146±0.059 | 0.180±0.039 | 0.281±0.084 | 0.395±0.079 | 0.512±0.156 |
| w.o. Multi-resolution | 0.048±0.013 | 0.137±0.030 | 0.159±0.025 | 0.276±0.075 | 0.387±0.076 | 0.540±0.162 |
| w. All blocks | **0.048±0.008** | **0.109±0.008** | **0.278±0.042** | **0.271±0.054** | **0.375±0.053** | **0.541±0.167** |

### 4.3. Ablation Study

We conducted a detailed ablation study to evaluate the effectiveness of each functional block, as is summarized in Table 2, Table 3 and Table 4. Experimental results in Table 2 and Table 3 demonstrate that the incorporation of GAT-Transformer blocks and multi-resolution information compensates for the limited details in the original bin-level data, yielding a PCC improvement of 0.079 on our dataset and 0.026 on the CRC dataset at $16\mu$m bins. At $8\mu$m bins, PCC increases by 0.098 and 0.029 on the VUMC and CRC datasets, respectively. Additionally, the consistency loss enhances the synergy of multi-resolution information, thereby facilitating more effective learning of high-resolution features and further improving the model's performance.

We also investigate the pathology-specific foundation model UNI (Chen et al., 2024) as the encoder for MagNet, with results summarized in Table 4. Compared with ResNet50, replacing it with UNI led to a slight decline in performance. An explanation is that the larger model size of UNI constrained the subgraph dimensions during training. To optimize computational efficiency, we process bin-level subgraphs iteratively, where the batch size determines the graph size. Under the same experimental conditions (one NVIDIA RTX A6000 GPU with 48GB memory), UNI's larger parameter count resulted in a reduced batch size to 64, compared with 256 for ResNet50. This reduction in subgraph size limited the model's ability to capture sufficient contextual information from neighboring bins, ultimately leading to the observed performance degradation.

Table 4: **Ablation study on backbone selection for MagNet.**

| Resolution | Backbone | VUMC (in-house dataset) | | | CRC (Oliveira et al., 2024) | | |
|---|---|---|---|---|---|---|---|
| | | MSE | MAE | PCC | MSE | MAE | PCC |
| $8\mu$m/112px | ResNet50 | 0.048±0.008 | **0.109±0.008** | **0.278±0.042** | **0.271±0.054** | **0.375±0.053** | **0.541±0.167** |
| | UNI | **0.047±0.006** | 0.112±0.003 | 0.266±0.049 | 0.331±0.088 | 0.422±0.079 | 0.505±0.142 |
| $16\mu$m/112px | ResNet50 | **0.127±0.024** | **0.228±0.034** | **0.378±0.057** | **0.564±0.184** | **0.581±0.114** | **0.574±0.154** |
| | UNI | 0.131±0.022 | 0.239±0.037 | 0.364±0.054 | 0.638±0.181 | 0.617±0.116 | 0.556±0.112 |
| $55\mu$m/224px | ResNet50 | **0.324±0.044** | **0.458±0.030** | **0.611±0.082** | **0.688±0.149** | **0.612±0.069** | **0.670±0.059** |
| | UNI | 0.339±0.038 | 0.469±0.024 | 0.582±0.044 | 0.735±0.231 | 0.638±0.084 | 0.643±0.111 |

### 5. Conclusion

We introduce a novel framework specifically tailored for high-resolution gene expression tasks. Our MagNet model integrates multi-level information and leverages spatial relationships derived from pathological images, effectively overcoming the input information bottleneck in HD gene expression prediction. Consequently, MagNet can accurately capture gene expression patterns at an 8 $\mu$m single-cell resolution. In addition, we present the first systematic and comprehensive evaluation of HD-level spatial transcriptomics datasets. We benchmarked MagNet against current state-of-the-art methods on two HD datasets under three different resolution settings. Experimental results demonstrate that MagNet consistently achieves top-tier predictive performance across multiple resolutions in both datasets. By extending gene prediction from the spot level to the cellular scale, MagNet establishes a new paradigm and benchmark for future research in spatial transcriptomics.

## Acknowledgments

This research was supported by NIH R01DK135597 (Huo), DoD HT9425-23-1-0003 (HCY), and KPMP Glue Grant. This work was also supported by Vanderbilt Seed Success Grant, Vanderbilt Discovery Grant, and VISE Seed Grant. This project was supported by The Leona M. and Harry B. Helmsley Charitable Trust grant G-1903-03793 and G-2103-05128. This research was also supported by NIH grants R01EB033385, R01DK132338, REB017230, R01MH125931, and NSF 2040462. We extend gratitude to NVIDIA for their support by means of the NVIDIA hardware grant. This work was also supported by NSF NAIRR Pilot Award NAIRR240055.

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

## Appendix A. Gene Selection and Estimation

To estimate the gene expression at the spot level and the region level, we aggregated the value of gene expression of 16 $\mu m$ bins within their respective spot and region areas. This process can be defined as:

$$y_s = \sum_{i \in S} y_i, \quad y_r = \sum_{i \in R} y_i \tag{6}$$

Here, $y_i$ denotes the gene expression value at the $i$-th bin, $S$ represents the set of bins within a specific spot, and $R$ denotes the set of bins within a certain area, thus ensuring the consistency of gene expression across multiple resolutions. The selected genes with the highest average expression for each dataset and resolution are presented in Figure. 4

| Dataset (Resolution) | Genes to be predicted |
|---|---|
| VUMC data (8$\mu m$) | 'EIF3A', 'RTN3', 'LGALS3BP', 'TIMP3', 'CCNL2', 'DSP', 'SLC38A1', 'MAT2A', 'VASP', 'STAU1', 'ACTN1', 'YBX3', 'ETS2', 'TTC3', 'CALM2', 'LUM', 'MBD2', 'KTN1', 'CRIP1', 'HMGCS2', 'C19orf33', 'PERP', 'CEACAM7', 'VMP1', 'RHOB', 'FAM3D', 'RNF7', 'ASPH', 'CD151', 'RHOC', 'KLF6', 'RBM3', 'SLC44A4', 'TGFBR2', 'SSR2', 'MIF', 'PDHA1', 'CA2', 'RAB10', 'FLNA', 'GPX4', 'MISP', 'C1S', 'XBP1', 'COL6A3', 'ZFP36', 'PSMA7', 'MYC', 'NBL1', 'CLDN7', 'TIMP1', 'JUND', 'DAB2', 'STK24', 'PTBP3', 'ACSL5', 'IGFBP7', 'ATP1A1', 'SELENOH', 'PSME1', 'GSTP1', 'RRBP1', 'SERPINB1', 'TFRC', 'RHOA', 'CAMK2N1', 'EIF4A1', 'IFITM3', 'PI3', 'MUC5B', 'TMEM176B', 'PNRC1', 'CTTN', 'MGP', 'HSPB1', 'HIST1H1B', 'GLUL', 'NPC2', 'APLP2', 'MYH14', 'C15orf48', 'SLC40A1', 'KRT20', 'TSC22D3', 'COL6A1', 'PRDX2', 'LGALS1', 'TOMM7', 'DDX5', 'TMEM106C', 'SOD2', 'TPM4', 'SRGN', 'TFF3', 'PDCD4', 'ZG16', 'CLDN4', 'CD44', 'CTSZ', 'PSAP', 'SLC12A2', 'HSD11B2', 'NDUFA4', 'MGAT4B', 'ITGA6', 'LAMP1', 'S100A11', 'ATP1B1', 'ANXA11', 'CIRBP', 'GPX2', 'ID1', 'EIF4EBP2', 'AGR2', 'DUSP1', 'LYZ', 'UQCRC1', 'DCN', 'ANXA2', 'CD63', 'GOLM1', 'AHNAK', 'CTSA', 'PLA2G2A', 'PRSS23', 'HSP90AB1', 'HNRNPA3', 'SH3BGRL3', 'KIF5B', 'STT3B', 'IFI27', 'STARD10', 'TPM2', 'SAT1', 'LMNA', 'CCNI', 'C3', 'MUC13', 'EGR1', 'TSPAN1', 'HIST1H1C', 'BRI3', 'ERGIC3', 'CALD1', 'MYL6', 'HSP90AA1', 'ELF3', 'COL6A2', 'SELENBP1', 'JUNB', 'TM9SF3', 'YBX1', 'HNRNPA2B1', 'A2M', 'MMP2', 'ACTG2', 'CTNND1', 'MUC2', 'CHCHD2', 'SPINT2', 'JUN', 'TSPAN3', 'YWHAZ', 'TM9SF2', 'EEF1B2', 'MYH11', 'KLF5', 'CLCA1', 'NME2', 'IGLC1', 'FXYD3', 'ZFP36L2', 'FOS', 'CTSB', 'CSRP1', 'SLC26A2', 'CKB', 'ST14', 'SELENOW', 'H3F3A', 'MCL1', 'DSTN', 'TXNIP', 'SPARC', 'PPDPF', 'EEF2', 'APP', 'SYNGR2', 'TXNDC5', 'HSPA8', 'ACTG1', 'CD9', 'MYL9', 'UBA52', 'LGALS4', 'PFN1', 'CFL1', 'PYGB', 'SELENOP', 'MT-ND6', 'CST3', 'CD24', 'FABP1', 'IGHM', 'CD74', 'TPM1', 'COL1A2', 'S100A10', 'LGALS3', 'DES', 'EEF1G', 'MT-ND5', 'LCN2', 'TAGLN', 'ITM2C', 'FTL', 'OLFM4', 'FCGBP', 'CEACAM6', 'KRT8', 'SLC26A3', 'IGHA1', 'OAZ1', 'COL1A1', 'TSPAN8', 'S100A6', 'B2M', 'EPCAM', 'VIM', 'MT-ND1', 'PHGR1', 'CEACAM5', 'TMSB10', 'COL3A1', 'ACTB', 'FTH1', 'MUC12', 'IGHG1', 'MT-ND2', 'JCHAIN', 'PIGR', 'MT-ND4L', 'MT-ND3', 'TMSB4X', 'MT-CYB', 'MT-CO2', 'MT-ND4', 'MT-CO3', 'MT-ATP6', 'IGKC' |
| VUMC data (16 & 55$\mu m$) | 'NDUFB9', 'BEX3', 'BSG', 'LRPAP1', 'CD9', 'LDHB', 'MSN', 'OGDHL', 'OCIAD2', 'SYNE2', 'CD63', 'EPS8L2', 'NPR3', 'CFDP1', 'COL4A1', 'GSTK1', 'NPC2', 'GABARAP', 'TPP1', 'WDR72', 'PRKAR1A', 'C12orf75', 'CFI', 'EID1', 'HIPK3', 'AKR1A1', 'RAB5C', 'JUNB', 'GNS', 'G6PC', 'CTSA', 'GLYATL1', 'TXNIP', 'OGA', 'PBLD', 'OPTN', 'RBPMS', 'EZR', 'CKB', 'AHNAK', 'PDK4', 'NDUFA1', 'PAX8', 'HIPK2', 'RHOC', 'UQCRB', 'SLC5A12', 'PPDPF', 'AGMAT', 'PDE4D', 'SFRP1', 'NDUFB5', 'ALDH2', 'TSPAN3', 'MMP7', 'COL4A2', 'ACY1', 'AQP2', 'PRODH2', 'MAF', 'IGFBP5', 'CAMK2N1', 'RNF152', 'ATP1A1', 'COL3A1', 'FAM107B', 'HSP90AA1', 'SLC16A9', 'LGMN', 'SDC4', 'EPCAM', 'PTGER3', 'PRDX2', 'CTTN', 'GSTP1', 'JUND', 'PRXL2A', 'S100A2', 'MT2A', 'ANXA11', 'RAB11FIP3', 'CRYL1', 'BBOX1', 'SHMT1', 'PPARGC1A', 'SLC22A2', 'C11orf96', 'FBP1', 'CTSH', 'APLP2', 'ZFP36L1', 'GLYAT', 'MMP24OS', 'PFN1', 'SLC7A8', 'CMBL', 'SPARC', 'ACTN4', 'SELENOW', 'CCNL2', 'UBA52', 'ECH1', 'CAPN2', 'CEBPD', 'NFE2L1', 'EEF1G', 'IGFBP4', 'DDX5', 'GALNT11', 'TMEM176A', 'FABP1', 'ERRFI1', 'CCN2', 'MAT2A', 'LMNA', 'EPAS1', 'SLC34A1', 'TMEM176B', 'KCNJ16', 'CALD1', 'ACAA1', 'PDZK1IP1', 'ALDH6A1', 'SLC8A1', 'JUN', 'SERPINA1', 'GDF15', 'DAB2', 'COL18A1', 'MCL1', 'SERF2', 'NUCB1', 'FXYD2', 'LENG8', 'CSRP1', 'SLC7A7', 'NDUFA4', 'LAMP1', 'DSTN', 'DPEP1', 'ISCU', 'SLC22A8', 'HSPB1', 'MT1G', 'HNRNPA2B1', 'WSB1', 'CDH16', 'PTH1R', 'SUCLG1', 'SLC4A4', 'DUSP1', 'CST3', 'BCAM', 'SLC3A2', 'SLC13A3', 'PCSK1N', 'SQSTM1', 'WFDC2', 'HSPA8', 'ALB', 'C11orf54', 'GPX4', 'VIM', 'CNDP2', 'AIF1L', 'MME', 'BRI3', 'PCK1', 'RBP5', 'FOS', 'MIOX', 'CGNL1', 'BHLHE40', 'MYL6', 'CLU', 'CCN1', 'NR4A1', 'EIF4EBP2', 'ADI1', 'TMSB10', 'CNN3', 'PEBP1', 'MT-ND5', 'PAH', 'CALB1', 'MGP', 'EGR1', 'C7', 'A2M', 'MT-ND6', 'PSAP', 'S100A6', 'LRP2', 'ENPEP', 'HPD', 'CCNI', 'ZFP36', 'BHMT2', 'TPM1', 'APP', 'RHOB', 'CD24', 'PIGR', 'CXCL14', 'SLC12A1', 'MT-ND2', 'CYB5A', 'MAL', 'TAGLN', 'ECHS1', 'ANPEP', 'CIRBP', 'OAZ1', 'CUBN', 'CRYAB', 'SMIM24', 'CA12', 'ITM2B', 'CLIC4', 'MT-ND1', 'CTSB', 'SLC12A3', 'EEF2', 'ASS1', 'FTL', 'ACTB', 'UMOD', 'SELENOP', 'BHMT', 'SAT1', 'AQP1', 'SLC3A1', 'DEFB1', 'CD74', 'FTH1', 'MT-ND3', 'ATP1B1', 'SPP1', 'IGFBP7', 'MT-ND4', 'GATM', 'MT-CO2', 'MT-ATP6', 'TMSB4X', 'MT-ND4L', 'MT-CYB', 'ALDOB', 'APOE', 'MT-CO3', 'GPX3' |
| CRC data (8$\mu m$) | 'CCNL2', 'MAT2A', 'RTN3', 'BSG', 'KTN1', 'SLC38A1', 'VASP', 'DSP', 'TRAM1', 'LGALS3BP', 'CRIP1', 'STAU1', 'YBX3', 'TTC3', 'CALM2', 'ETS2', 'LUM', 'FAM3D', 'RHOB', 'C19orf33', 'MBD2', 'HMGCS2', 'VMP1', 'CEACAM7', 'PERP', 'ASPH', 'RNF7', 'CD151', 'RHOC', 'TGFBR2', 'KLF6', 'RBM3', 'SSR2', 'TIMP1', 'FLNA', 'RAB10', 'SLC44A4', 'MIF', 'C1S', 'XBP1', 'PDHA1', 'DAB2', 'MYC', 'MISP', 'GPX4', 'COL6A3', 'ZFP36', 'PTBP3', 'CLDN7', 'NBL1', 'CA2', 'IGFBP7', 'PSMA7', 'STK24', 'JUND', 'ACSL5', 'SELENOH', 'ATP1A1', 'IGHG3', 'GSTP1', 'PSME1', 'RRBP1', 'SERPINB1', 'CAMK2N1', 'TFRC', 'RHOA', 'EIF4A1', 'MUC5B', 'IFITM3', 'MGP', 'PNRC1', 'TMEM176B', 'CTTN', 'GLUL', 'HSPB1', 'PI3', 'NPC2', 'HIST1H1B', 'APLP2', 'MYH14', 'PRDX2', 'COL6A1', 'SLC40A1', 'LGALS1', 'C15orf48', 'TSC22D3', 'KRT20', 'TOMM7', 'TPM4', 'TMEM106C', 'DDX5', 'SOD2', 'PDCD4', 'ZG16', 'CLDN4', 'SRGN', 'TFF3', 'NDUFA4', 'PSAP', 'SLC12A2', 'CTSZ', 'CD44', 'ITGA6', 'ID1', 'S100A11', 'MGAT4B', 'ANXA11', 'HSD11B2', 'LAMP1', 'CIRBP', 'EIF4EBP2', 'GPX2', 'ATP1B1', 'DUSP1', 'UQCRC1', 'DCN', 'LYZ', 'CD63', 'AGR2', 'GOLM1', 'ANXA2', 'PLA2G2A', 'PRSS23', 'AHNAK', 'CTSA', 'STT3B', 'HNRNPA3', 'SH3BGRL3', 'HSP90AB1', 'KIF5B', 'IFI27', 'STARD10', 'LMNA', 'TPM2', 'HIST1H1C', 'CCNI', 'SAT1', 'C3', 'EGR1', 'BRI3', 'ERGIC3', 'CALD1', 'TSPAN1', 'MUC13', 'MYL6', 'HSP90AA1', 'ELF3', 'COL6A2', 'JUNB', 'SELENBP1', 'TM9SF3', 'YBX1', 'A2M', 'HNRNPA2B1', 'ACTG2', 'MUC2', 'CTNND1', 'MMP2', 'JUN', 'CHCHD2', 'SPINT2', 'YWHAZ', 'TSPAN3', 'TM9SF2', 'MYH11', 'EEF1B2', 'KLF5', 'CLCA1', 'NME2', 'FOS', 'ZFP36L2', 'FXYD3', 'CTSB', 'CSRP1', 'IGLC1', 'SLC26A2', 'CKB', 'ST14', 'SELENOW', 'H3F3A', 'DSTN', 'MCL1', 'TXNIP', 'SPARC', 'EEF2', 'HSPA8', 'SYNGR2', 'APP', 'PPDPF', 'ACTG1', 'TXNDC5', 'MYL9', 'CD9', 'UBA52', 'LGALS4', 'PFN1', 'PYGB', 'CFL1', 'SELENOP', 'CST3', 'MT-ND6', 'CD24', 'FABP1', 'TPM1', 'CD74', 'COL1A2', 'S100A10', 'DES', 'IGHM', 'LGALS3', 'EEF1G', 'MT-ND5', 'LCN2', 'TAGLN', 'ITM2C', 'FTL', 'OLFM4', 'FCGBP', 'CEACAM6', 'KRT8', 'SLC26A3', 'OAZ1', 'COL1A1', 'IGHA1', 'TSPAN8', 'S100A6', 'B2M', 'EPCAM', 'VIM', 'MT-ND1', 'TMSB10', 'CEACAM5', 'PHGR1', 'COL3A1', 'ACTB', 'FTH1', 'MUC12', 'IGHG1', 'MT-ND2', 'JCHAIN', 'PIGR', 'MT-ND4L', 'MT-ND3', 'TMSB4X', 'MT-CYB', 'MT-CO2', 'MT-ND4', 'MT-CO3', 'MT-ATP6', 'IGKC' |
| CRC data (16 & 55$\mu m$) | 'MSN', 'CLTRN', 'CFDP1', 'CD9', 'BSG', 'CD63', 'DYNLL2', 'NPR3', 'DCN', 'SYNE2', 'CYS1', 'OGDHL', 'OCIAD2', 'JUNB', 'EPS8L2', 'GSTK1', 'C12orf75', 'WDR72', 'HIPK3', 'GLYATL1', 'CFI', 'GABARAP', 'AKR1A1', 'TPP1', 'NPC2', 'RAB5C', 'PRKAR1A', 'TXNIP', 'GNS', 'COL4A1', 'PBLD', 'EID1', 'CTSA', 'EZR', 'OGA', 'AHNAK', 'CKB', 'G6PC', 'OPTN', 'PDK4', 'RBPMS', 'NDUFA1', 'HIPK2', 'PAX8', 'RHOC', 'PPDPF', 'SFRP1', 'PDE4D', 'AGMAT', 'ALDH2', 'AQP2', 'CAMK2N1', 'PRODH2', 'MMP7', 'COL4A2', 'NDUFB5', 'UQCRB', 'TSPAN3', 'MAF', 'SLC5A12', 'FAM107B', 'SDC4', 'ACY1', 'COL3A1', 'LGMN', 'SLC16A9', 'RNF152', 'PRDX2', 'HSP90AA1', 'IGFBP5', 'EPCAM', 'SHMT1', 'ATP1A1', 'JUND', 'S100A2', 'PTGER3', 'GSTP1', 'CTTN', 'RAB11FIP3', 'CRYL1', 'ANXA11', 'MT2A', 'PPARGC1A', 'C11orf96', 'BBOX1', 'PRXL2A', 'SLC22A2', 'APLP2', 'CTSH', 'ZFP36L1', 'PFN1', 'CMBL', 'MMP24OS', 'GLYAT', 'CAPN2', 'FBP1', 'SLC7A8', 'SPARC', 'CCNL2', 'UBA52', 'GALNT11', 'NFE2L1', 'SELENOW', 'ECH1', 'ACTN4', 'ERRFI1', 'CEBPD', 'TMEM176A', 'DDX5', 'FABP1', 'EEF1G', 'IGFBP4', 'MAT2A', 'TMEM176B', 'CCN2', 'SLC8A1', 'EPAS1', 'LMNA', 'COL18A1', 'SLC34A1', 'ACAA1', 'GDF15', 'JUN', 'KCNJ16', 'FXYD2', 'ALDH6A1', 'CALD1', 'SERPINA1', 'DAB2', 'MCL1', 'NDUFA4', 'SLC7A7', 'SERF2', 'PDZK1IP1', 'NUCB1', 'HSPB1', 'DPEP1', 'WSB1', 'LAMP1', 'HNRNPA2B1', 'CSRP1', 'SLC22A8', 'PTH1R', 'ISCU', 'DSTN', 'LENG8', 'MT1G', 'CST3', 'CDH16', 'DUSP1', 'SLC4A4', 'BCAM', 'PCSK1N', 'SLC3A2', 'SUCLG1', 'WFDC2', 'SQSTM1', 'SLC13A3', 'HSPA8', 'VIM', 'GPX4', 'C11orf54', 'ALB', 'CNDP2', 'MME', 'AIF1L', 'RBP5', 'FOS', 'PCK1', 'BRI3', 'BHLHE40', 'MIOX', 'MYL6', 'CLU', 'CCN1', 'CGNL1', 'CNN3', 'NR4A1', 'EIF4EBP2', 'ADI1', 'PEBP1', 'TMSB10', 'CALB1', 'PAH', 'MT-ND5', 'C7', 'EGR1', 'MGP', 'A2M', 'S100A6', 'MT-ND6', 'PSAP', 'LRP2', 'ZFP36', 'CCNI', 'ENPEP', 'HPD', 'BHMT2', 'TPM1', 'APP', 'RHOB', 'CD24', 'PIGR', 'CXCL14', 'SLC12A1', 'MT-ND2', 'MAL', 'CYB5A', 'TAGLN', 'ANPEP', 'ECHS1', 'CIRBP', 'CUBN', 'OAZ1', 'SMIM24', 'ITM2B', 'CRYAB', 'CA12', 'CLIC4', 'MT-ND1', 'CTSB', 'EEF2', 'ASS1', 'FTL', 'SLC12A3', 'ACTB', 'UMOD', 'SELENOP', 'BHMT', 'SAT1', 'AQP1', 'SLC3A1', 'CD74', 'DEFB1', 'FTH1', 'MT-ND3', 'ATP1B1', 'SPP1', 'IGFBP7', 'MT-ND4', 'GATM', 'MT-CO2', 'MT-ATP6', 'TMSB4X', 'MT-ND4L', 'MT-CYB', 'ALDOB', 'APOE', 'MT-CO3', 'GPX3' |

Figure 4: **Gene selection in each dataset and resolution.**

