# OpenReview forum: "MagNet: Multi-Level Attention Graph Network for Predicting High-Resolution Spatial Transcriptomics"
_MIDL.io/2025/Conference — MIDL 2025 Oral_

### Official Review · Reviewer_udUS · 2025-02-15

**Confidence:** 5
**Preliminary Rating:** 4
**Recommendation:** Poster
**Final Rating:** 4

**Summary:**

This paper introduces MagNet, a multi-level attention graph network to predict high-resolution spatial transcriptomics. The key idea of the paper is to use histology and transcriptomics data at different scales (8um bin level, 56um spot level, and 128um region level) to achieve high-resolution prediction. The authors have tested their models on two datasets; one private dataset of Visium HD on kidney tissues and another public dataset provided by 10X Geomics Visium HD on colorectal cancer tissues. The most significance of the model is that such high-resolution prediction cannot be achieved by previous prediction models that aim on low-resolution prediction on the spot level. Knowing the high-resolution spatial transcriptomics can inform scientists the spatial cellular structures in a tissue.

**Strengths:**

- The idea of integrating histology and transcriptomics at different scales is interesting at this time. Even though this paper is not the first one that proposes such idea (at least TRIPLEX used the same idea before), the paper introduces a new way of implementation of such ideas. In particular, the consistency loss between predictions at different scales and the cross-attention layer mixing histology embeddings across scales make it possible for the model to share information from local area to global area.

- The comprehensive experiments for five previous prediction models are useful for other researchers. At least to my knowledge, no one has done the experiments across different resolutions. It is enough interesting to see how these models perform under different resolutions.

- It will be valuable if the authors can share the private kidney dataset. Making the dataset public not only enables others to verify the result, but also benefits the community of computational biology to improve such prediction models, either Magnet or other models.

**Weaknesses:**

- MagNet does not address the problem of batch effects in histology images. It is known that H&E stained histology images from different medical centers have very different colors. There is not a particular module in the model addressing this concern. The experiment data, either the kidney dataset (VUMC) or the colorectal cancer (CRC), is coming from the same lab and thus should not have any substantial batch effects on histology. At least the four colorectal samples in 10X Genomics CRC dataset do not have any batch effects on histology.

- MagNet requires training data at single-cell or nearly sing-cell levels spatial data. Currently these technology such as Visium HD, Xenium, CoxMx are expensive and the training data is very few.

- In the experiment, the authors should include comparison with TRIPLEX, since it is the newest prediction model that uses the same idea of leveraging information across scales.
    - It is worth noting that the PCC values for ST-Net, EGN, HisToGene, BLEEP, and Hist2ST at 55um level are inflating. These previous models typically have 0.2-0.3 PCC on 55um spot level (for example, see BLEEP Table 1) . However, the experiment in this paper reports 0.6-0.7 PCC. One reason is that the dataset used in the experiment is more homogeneous than the dataset used in the papers of previous models, thus making the task much easier.

- The hyperparameters on the loss functions are not easy to tune. The authors do not claim how they tune the hyperparemeters. If the authors tune them on the same dataset of the experiment, then there will be a valid concern of overfitting.

-  Random selecting bins and spots for training and testing is not a common way in similar prediction task. The authors claim in Section 3 Data Prepocessing that they random selected 6,000 bins and 2,500 spot-level patches. They do not justify why they do not use all the bins and spots inside the tissue. Moreover, the author do not specify how many global-level patches they selected.

**Detailed Comments:**

- Figure 3 is not very informative. It is better to illustrate the predictions on the whole slide for some marker genes between MagNet and the best competing prediction models.

- MSE is sometimes misleading. Since many genes are sparsely distributed, an underestimated model gives a lower MSE (or MAE) than an overestimated model. Therefore, it is more informative to see the prediction on the whole slide, instead of looking at numerical values or zooming into a single region.

- It is not known if the authors separate dataset into training and testing by patients or by samples or by bins. If a patient's data is present in both training and testing data, the prediction task is then easier.

**Justification Of The Final Rating:**

I will maintain my previous rating at Weak accept.

I agree with Reviewer Pp8q that the visualization of prediction at WSI level is not informative. It looks like the expression is constant over the entire tissue. This seems like a mistake made when the authors plotted the figures.

**Justification Of The Preliminary Rating:**

Overall, this paper presents an interesting framework of using multi-scale information to do prediction. There is few predition models at high-resolution. Even though MagNet does not have significant increases in prediction accuracy than previous models, this paper is still a good implementation of the idea. Moreover, the experiment tabel for previous models give new information on the performance of these models on different prediction scales. This table is worthwhile to be published.

**Questions To Address In The Rebuttal:**

- Could the authors add a comparison between TRIPLEX and MagNet?

- How do the authors tune the hyperparameters? Do they used the same experiment data to tune?

- Can you give more illustrations on the gene prediction on the whole slide? Just like what BLEEP or TRIPLEX did in their papers.

**Special Issue:**

No

---

> ### Author Response · Authors · 2025-03-07
> **Official Comments for Reviewer udUS**
>
> **WSI-level Performance:** We appreciate this suggestion and have incorporated WSI-level performance in our revised version and expanded Figure 3 to include WSI-level SGPP1 prediction (Figure 3B). Additionally, we evaluated another kidney-related gene, DPEP1, on different samples in our VUMC data, with results visualized in Figure 6 in Appendix B. MagNet achieved MSEs of 0.0544 / 0.0493 for SGPP1 / DPEP1 at the WSI level, significantly outperforming other methods such as EGN (0.1605 / 0.1855) and BLEEP (0.1530 / 0.1126), further demonstrating its superiority in HD-level gene expression prediction.
>
> **Train-Test Partitioning:** We adopted different data separation strategies for the VUMC and CRC datasets due to our dataset’s limited size. For CRC dataset (4 samples, 4 WSIs), each fold in cross-validation used one WSI for validation and the other three for training. In our dataset (2 samples, 12WSIs), each fold used one full sample (six WSIs) plus half of the second sample (three WSIs) for training, with the remaining half for validation, thus maximizing patient-level data partitioning despite limited sample size.
>
> **Data selection:** HD-level bins are extremely large in number, with approximately 150,000 at 16μm and millions at 8μm per WSI, compared with just thousands at spot level. Additionally, significant overlap between paired patches of adjacent bins leads to resource redundancy during training. To mitigate this, we use a sparse sampling strategy, randomly selecting bins and spots to sufficiently cover the WSI for training and testing. For Global (Region)-level patches, we crop the WSI into 512×512 patches and determine pairing across resolutions based on overlap. Each WSI typically contains 1,000–2,000 region-level patches.
>
> **Hyperparameter Fine-tuning:** We use cross-validation for hyperparameter tuning. Given the limited dataset, cross-validation allows the model to be trained and validated multiple times across different data splits, reducing biases caused by specific partitioning, thus ensuring the generalization of hyperparameter tuning while preventing overfitting to a particular training set.
>
> **Batch effect and Our Motivation:** Removing batch effects remains a challenge in computational pathology, but recent methods such as canonical correlation analysis [1] and mutual nearest neighbors [2] have made progress. Our work is orthogonal yet complementary to these efforts by offering a novel framework and establishing the first paradigm and benchmark for HD single-cell gene expression prediction.
>
> **Comparison with TRIPLEX:** We appreciate this valuable suggestion and have incorporated the experimental results of TRIPLEX in Table 1.
>
> [1] Butler, A., Hoffman, P., Smibert, P., Papalexi, E. & Satija, R. Integrating single-cell transcriptomic data across different conditions, technologies, and species. *Nat. Biotechnol.* **36**, 411–420 (2018).
>
> [2] Haghverdi, L., Lun, A. T. L., Morgan, M. D. & Marioni, J. C. Batch effects in single-cell RNA-sequencing data are corrected by matching mutual nearest neighbors. *Nat. Biotechnol.* **36**, 421–427 (2018).

---

> > ### Comment · Reviewer_udUS · 2025-03-07
> >
> > I am satisfied with the authors' response. They have updated the comparison with TRIPLEX and gave more clarifications to hyperparameter tuning and train-test splitting.
> >
> > **WSI-level Performance**: In the updated Figure 3B, 5, and 6, I cannot see a specific pattern for these genes. It is hard to tell visually how good the prediction is. If the authors still have time to revise, consider plotting the predicted gene expression and make the visualization clearer. A reference plot would be Figure 2 in the TRIPLEX paper or Figure 3 in the BLEEP paper.
> >
> > **Train-Test Partitioning**: The authors' data splitting method for the CRC dataset splits data into training and testing by patients. However, the authors' data splitting method for their own dataset makes the training and test data contain the same patient. It is fine in the benchmark part since all methods are using the same splitting protocols. However, it should be alarmed that in practice, it is highly unlikely that the model trained will contain the new patient's data.
> >
> > **Hyperparameter Fine-tuning**: I'm okay with the authors' way of hyperparameter tunning since there are limited data. However, since the authors didn't have a separate test data (never used in hyperparameter tunning), the authors' hyperparameter tuning will lead to a overfitting model.
> >
> > The authors may consider their own availability to decide whether they want to revise the paper in the next few days or not. I am happy with the current version of the paper.

---

> > > ### Author Response · Authors · 2025-03-12
> > > **Response to Reviewer udUS Final Comments**
> > >
> > > Thank you for recognizing our efforts and recommending acceptance of our manuscript. We appreciate your constructive comments and would like to further clarify your remaining concerns as follows:
> > >
> > > **WSI-level Visualization Concerns:** The observed visual appearance of constant gene expression at the WSI-level primarily results from two factors: (1) Gene sparsity inherent to HD-level data, and (2) High bin density within WSIs.
> > >
> > > - **Gene sparsity:**
> > >    The tissue area captured by HD-level data is significantly smaller compared to conventional spot-level spatial transcriptomics (approximately 55 µm diameter per spot). Consequently, the amount of captured gene expression per bin is notably reduced—approximately 1/16 for 16 µm bins and as low as 1/64 for 8µm bins—leading to uniformly low expression of most genes across the vast majority of bins. Even after selecting the 250 most highly expressed genes, this sparsity contributes to the visually constant appearance observed in the visualization results.
> > > - **Bin density:**
> > >    Compared with traditional spot-level data, which typically include only hundreds to thousands of spots per WSI, HD-level analyses significantly increase the density of bins to achieve higher spatial resolution. A single WSI commonly comprises tens of thousands of bins at 16µm resolution and over one million bins at 8µm resolution. The extremely high density of bins, coupled with the image compression during visualization, further enhances the impression of visual constancy when gene-expression predictions are displayed at the WSI level.
> > >
> > > Following your suggestion, we have further improved our WSI-level gene expression visualizations by referring to the visualization strategies employed in *TRIPLEX* and *BLEEP*. These improved visualizations will be included in the final version of our manuscript.

---

### Official Review · Reviewer_Pp8q · 2025-02-22

**Confidence:** 4
**Preliminary Rating:** 3
**Recommendation:** Poster
**Final Rating:** 4

**Summary:**

The authors study the problem of predicting high-resolution spatially-resolved gene-expression from pathology whole-slide images (WSIs).

They propose MagNet, a model that utilizes cross-attention layers to integrate features from multi-resolution image patches, and graph integration blocks for spatial modelling.

They conduct experiments on an in-house kidney dataset of 12 HD ST samples, and a public colorectal dataset of 4 samples. Under three different resolution settings for the predictions.

They compare the performance of the proposed MagNet with 5 baseline models. Using 4-fold cross-validation on the WSI-level in both datasets. MagNet seems to perform well compared to the baselines.

**Strengths:**

- The studied problem is interesting and relevant.

- This is a very well written paper overall with nice-looking and illustrative figures etc.

- The proposed MagNet model conceptually makes sense overall, and seems to perform well compared to relevant baselines.

**Weaknesses:**

- The experiments are conducted only on two different datasets, with just 16 WSIs in total.

- Only cross-validation results, no external validation.

- Given the two above points, I think the authors oversell the significance/strength of their results in Section 4. The proposed MagNet is optimized for the two considered datasets (Section 4.3), these datasets are not particularity big, and only cross-validation results are presented. "_the relatively low standard deviation observed among all metrics during cross-validation highlights the method’s robustness and stability, underscoring its reliability for practical clinical applications_" is definitely a too strong claim in my opinion. Also, in Section 4.2, the results presented are for a single gene on a single WSI?

**Detailed Comments:**

Questions/suggestions:
- In Section 2.1 you write "bin, spot and region levels", but in Figure 2 you have bin-level, spot-level and global-level. Should consistently use either "global" or "region"? "Global" should be changed to "Region" in Figure 2?

- Figure 2 is great, but Section 2.1 I think could be a bit more clear. Using eq. (1) you get $f^{'s}_b$ and $f^{'r}_b$, and then these two vectors are concatenated with $f_b$ and fed to the graph construction step in the middle of Figure 2?

- It would be interesting to compare the performance obtained with the current approach (optimizing a Resnet50 image encoder) with using a frozen pathology-specific foundation model (e.g. UNI), that could perhaps improve the prediction performance further?

- It could perhaps be possible to utilize the [HEST-1k](https://github.com/mahmoodlab/HEST) dataset for evaluating the trained models on some external data?




Minor things:
- "Zhu" at the top of even-numbered pages looks off-centered? Also, should be "Zhu et al."?

- Section 2.3, "and Pearson Correlation Coefficient loss (P)" --> "and Pearson Correlation Coefficient loss (PCC)"? Also change "P" to "PCC" in eq. (3) and (4)?

- Section 4.1, "We conducted four-fold validation on the WSI level", validation --> cross-validation?

**Justification Of The Final Rating:**

I like this paper overall, it is well written and the studied problem is interesting. The proposed MagNet model conceptually makes sense, and seems to perform well compared to relevant baselines. The experimental evaluation is however limited to two relatively small datasets, with just cross-validation results. But overall, I think the paper deserves to be accepted.

**Justification Of The Preliminary Rating:**

I like this paper overall, it is well written and the studied problem is interesting. I want to be able to recommend accept.

However, I think that the claims regarding the model performance in Section 4 need to be toned down in the current version.

**Questions To Address In The Rebuttal:**

See Weaknesses and Questions/suggestions above.

While additional experiments would be really nice, I understand that you're limited on space. The main thing I think should be addressed in the rebuttal/revision is just to tone down the claims in Section 4 a bit.

**Special Issue:**

No

---

> ### Author Response · Authors · 2025-03-07
> **Official Comments for Reviewer Pp8q**
>
> **Pathology-specific Backbone:** We appreciate this suggestion and have included its performance in Table 4. Compared with the original ResNet50 encoder, replacing it with UNI resulted in a slight performance decline. One possible explanation is that the model size constrained the subgraph dimensions during training. To minimize computational costs, we iteratively process bin-level subgraphs, where the graph size is determined by the batch size. Under the same experimental setup (one NVIDIA RTX A6000 GPU with 48GB memory), UNI's larger parameter size lead to a smaller batch size of 64, compared with 256 for ResNet50. The reduced subgraph size limited the model's ability to extract sufficient information from neighboring bins, ultimately leading to the observed performance drop.
>
> **External Validation:**  We appreciate this suggestion and have explored the HEST-1k dataset. However, the current HEST-1k dataset contains a limited number of HD samples (n=7) and includes only samples from Mouse Kidney, Mouse Embryo, and Human Pancreas. Since our models are specifically trained on human kidney and CRC, they are not suitable for cross-species or cross-organ evaluation.
>
> **Description in Section 2:** Features from all three levels are concatenated before undergoing the subsequent GAT-Transformer process. This operation enhances feature representation by effectively integrating information across different resolution levels. Additionally, we have updated Figure 2 and replaced "Global" with "Region" to ensure consistency throughout the context.
>
> **Minor Things:** We appreciate this suggestion and have corrected all typos and clarified figures in the revised version, with changes highlighted in blue text.

---

> > ### Comment · Reviewer_Pp8q · 2025-03-12
> >
> > Thank you for the response.
> >
> > I have read the other reviews and all rebuttals.
> >
> > The two other reviews are positive overall.
> >
> > The author responses are quite thorough. The updated paper includes a new baseline method, and two relevant ablation studies.
> >
> > The authors don't comment on my main concern _"The main thing I think should be addressed in the rebuttal/revision is just to tone down the claims in Section 4 a bit"_ in their response. While the claim at the end of Section 4.2 has been modified in the updated paper, _"Furthermore, the relatively low standard deviation observed among all metrics during cross-validation highlights the method’s robustness and stability, underscoring its reliability for practical clinical applications"_ is still at the end of Section 4.1. I still think this is a too strong claim.
> >
> > I'm somewhat confused about the WSI-level visualizations in the new Figure 3.b and Figure 5 - 6. Is the true and predicted gene-expression always just constant across the entire slide?
> >
> > Minor things:
> > - "Zhu" at the top of even-numbered pages looks off-centered? Also, should be "Zhu et al."?
> > - - This is still not fixed.
> > - Table 4, the MSE for VUMC, first row: UNI should be bold instead of Resnet50 here?
> >
> > If these things are addressed, I will increase my score to 4: Weak accept.

---

> > > ### Author Response · Authors · 2025-03-12
> > > **Response to Reviewer Pp8q**
> > >
> > > We sincerely appreciate your helpful feedback and the insightful suggestions on paper writing and WSI-level visualization. Below, we would like to clarify your remaining concerns further:
> > >
> > > **Claims in Section 4:** To address this concern, we have revised our wording at the end of Section 4.1. The final manuscript will be presented as follows: *"The relatively low standard deviation observed during cross-validation suggests stability of our MagNet under the tested conditions."*
> > >
> > > ---
> > >
> > > **WSI-level Visualization Concerns:** The observed visual appearance of constant gene expression at the WSI-level primarily results from two factors: (1) Gene sparsity inherent to HD-level data, and (2) High bin density within WSIs.
> > >
> > > - **Gene sparsity:**
> > >    The tissue area captured by HD-level data is significantly smaller compared to conventional spot-level spatial transcriptomics (approximately 55 µm diameter per spot). Consequently, the amount of captured gene expression per bin is notably reduced—approximately 1/16 for 16 µm bins and as low as 1/64 for 8µm bins—leading to uniformly low expression of most genes across the vast majority of bins. Even after selecting the 250 most highly expressed genes, this sparsity contributes to the visually constant appearance observed in the visualization results.
> > > - **Bin density:**
> > >    Compared with traditional spot-level data, which typically include only hundreds to thousands of spots per WSI, HD-level analyses significantly increase the density of bins to achieve higher spatial resolution. A single WSI commonly comprises tens of thousands of bins at 16µm resolution and over one million bins at 8µm resolution. The extremely high density of bins, coupled with the image compression during visualization, further enhances the impression of visual constancy when gene-expression predictions are displayed at the WSI level.
> > >
> > > To clearly demonstrate the differences between our model's predictions and the ground truth, we specifically zoomed into a small area and selectively sampled discrete bins for enhanced visualization, as shown in Figure 3A. Additionally, following Reviewer udUS's suggestion, we have further refined our WSI-level gene expression visualizations and will incorporate them into the final version of the manuscript.
> > >
> > > ---
> > >
> > > **Minor things:**  As recommended, we will address these writing issues in the final version by centering and updating the header on even-numbered pages to *"Zhu et al."*, and revising Table 4 accordingly.

---

> > > > ### Comment · Reviewer_Pp8q · 2025-03-14
> > > >
> > > > Thank you for the response.
> > > >
> > > > I will increase my score to 4: Weak accept.

---

### Official Review · Reviewer_rbCu · 2025-02-25

**Confidence:** 3
**Preliminary Rating:** 4
**Recommendation:** Poster
**Final Rating:** 5

**Summary:**

The authors propose a model for the prediction of spatial transcriptomics from high-resolution histopathological images. Importantly, the architecture integrates features from various resolution levels to do this. Additionally, at each level, the authors use a range of functional blocks and their usefulness is shown in an ablation study across levels. The method is compared against a range of baselines and it consistently performs best on both and open source and an in-house dataset.

**Strengths:**

- Whole transcriptome protocols such as the widely used 10x Visium v2 face resolution issues with spots containing expression data of many cells. While other architectures focus on such data, the authors propose a model that works particularly well on the high-resolution data Visium HD, tackling that issue in DL for spatial transcriptomics.
- The experiments and ablation seem sound and substantiate the authors' performance claims.
- Open-source code is provided.

**Weaknesses:**

- Some minor details regarding writing should be addressed, see below.
- A high-resolution-level-only variant could be added: If the authors wanted to prove their claim that the MagNet's multilevel feature aggregation leads to increased performance on various input resolutions, a baseline with only the highest resolution level could be added. This is a suggestion for the sake of completeness.
- Either Figure 3 needs introduction or more generally it should be introduced what the data looks like.

**Detailed Comments:**

Minor issues:
- "HD" is not introduced as Visium HD data
- "GAT" is not introduced
- Fig. 3 is referenced in the text as "Figure 4"
- The learning rate schedule is described as a cosine decay from 1e-4 down to 1e-2, which is an increase
- ST-Net's reference in the data preprocessing section leads to the ResNet paper.

**Justification Of The Final Rating:**

The authors addressed all points. Also, how they replied to the other reviews and adjusted the manuscript accordingly looks very decent. By adding experimental results, the manuscript's claims seem justified, the paper is clearly written and adds value to the field of DL in spatial transcriptomics.
(In Table 4 one bold value is mixed up, but I saw reviewer Pp8q noted this already.)

**Justification Of The Preliminary Rating:**

The manuscript addresses a relevant topic for MIDL, it is concise and the experiments good. The model outperforms nearly all baselines in nearly all resolutions on two datasets. With the above comments addressed, I'm confident that this work should be accepted.

**Questions To Address In The Rebuttal:**

The above comments should be addressed.

---

> ### Author Response · Authors · 2025-03-07
> **Official Comments for Reviewer rbCu**
>
> **High-resolution-level-only Baseline:** We appreciate this suggestion and have added a baseline using only the highest resolution (8μm) in Table 3. After incorporating multi-resolution features, the model achieved PCC improvements of 0.098 and 0.029 on the VUMC and CRC datasets, respectively. This result aligns with the findings from the ablation study on 16μm gene expression prediction, further demonstrating the necessity of multi-resolution feature integration.
>
> **Introduction of Figure 3:** The original Figure 3 presents randomly selected 16μm bins within a certain region in WSI, showing their distribution in predicting kidney-related gene SGPP1 expression and its MSE. Following R1’s suggestion, we expanded Figure 3 to include WSI-level SGPP1 prediction (Figure 3B).
>
> **Writing Issues**: We appreciate the suggestions and have corrected all writing issues in the revised version, with changes highlighted in blue text.

---

> > ### Comment · Area_Chair_1u4Y · 2025-03-14
> >
> > Dear reviewer, does this address your concerns?
> >
> > Your AC

---

> > > ### Comment · Reviewer_rbCu · 2025-03-14
> > >
> > > Yes, it does. I revised my review.

---

### Author Rebuttal · Authors · 2025-03-07

**Rebuttal:**

We sincerely thank all the reviewers and ACs for their constructive feedback. This rebuttal addresses the common concerns raised by the reviewers (R1: udUS, R2: Pp8q, R3: rbCu). Detailed point-by-point responses are provided below each reviewer’s comment.

**Supplementary Experiments (R1, R2, R3):** We appreciate the reviewers' suggestions and have incorporated supplementary results in the revised version. Added experiments include: (1) A comparison with TRIRLEX is presented in Table 1. (2) We adopt the pathology-specific foundation model UNI as the encoder, evaluate its performance in Table 4, and analyze the reasons for its slight performance decline in Section 4.3. (3) A baseline using only the highest resolution (8μm) is added in Table 3, demonstrating the necessity of multi-resolution feature integration.

**Batch Effect and External Data (R1, R2):** Removing batch effects remains a challenge in computational pathology, but recent methods such as canonical correlation analysis *(Butler et al., Nat. Biotechnol., 2018)* and mutual nearest neighbors *(Haghverdi et al., Nat. Biotechnol., 2018)* have made progress. Our work is orthogonal yet complementary to these efforts by offering a novel framework and establishing the first paradigm and benchmark for HD single-cell gene expression prediction. Meanwhile, the scarcity of HD data limits external validation, as even the currently most comprehensive HEST-1k dataset (as suggested by R2) lacks HD-level human kidney and CRC samples.

**Performance on WSIs (Figure 3) (R1, R2, R3):** The original Figure 3 presents randomly selected 16μm bins within a region in WSI, showing their distribution in predicting kidney-related gene SGPP1 expression and its MSE. Following R1’s suggestion, we expanded Figure 3 to include WSI-level SGPP1 prediction (Figure 3B). Additionally, we analyzed tubule-related gene DPEP1 and evaluated SGPP1 and DPEP1 prediction on WSIs from two samples in our VUMC dataset (see Figure 5 and Figure 6 in Appendix B). Results show that MagNet achieved MSEs of 0.0544 / 0.0493 for SGPP1 / DPEP1 at the WSI level, significantly outperforming other methods like EGN (0.1605 / 0.1855) and BLEEP (0.1530 / 0.1126), further validating its superiority in HD-level gene expression prediction.

**Typos and Figure Clarification (R1, R2, R3)**: We appreciate the suggestions and have corrected all typos and clarified figures in the revised version, with changes highlighted in blue text.

**Supporting Material:**

/attachment/f1c1d4e476d28d3f6755f9e906bf075f783f5108.pdf

---

### Meta-Review · Area_Chair_1u4Y · 2025-03-19

**Recommendation:** Accept (Poster)
**Confidence:** 5

**Metareview:**

The authors propose a new architecture to predict of spatial transcriptomics from high-resolution histopathological images, which makes use various resolution levels. The reviewers liked the overall idea and praised it as timely, they thought the experimental validation was rigorous. The authors addressed several concerns convinvingly during the rebuttal period. All reviewers are in favor of accepting.